# OVS Meets Continual Learning: Towards Sustainable Open-Vocabulary Segmentation

**Dongjun Hwang**[1]  **Yejin Kim**[1]  **Minyoung Lee**[1]  **Seong Joon Oh**[2,3]  **Junsuk Choe**[1†]

[1]Sogang University  [2]University of Tübingen  [3]Tübingen AI Center

## Abstract

Open-Vocabulary Segmentation (OVS) aims to segment classes that are not present in the training dataset. However, most existing studies assume that the training data is fixed in advance, overlooking more practical scenarios where new datasets are continuously collected over time. To address this, we first analyze how existing OVS models perform under such conditions. In this context, we explore several approaches such as retraining, fine-tuning, and continual learning but find that each of them has clear limitations. To address these issues, we propose ConOVS, a novel continual learning method based on a Mixture-of-Experts framework. ConOVS dynamically combines expert decoders based on the probability that an input sample belongs to the distribution of each incremental dataset. Through extensive experiments, we show that ConOVS consistently outperforms existing methods across pre-training, incremental, and zero-shot test datasets, effectively expanding the recognition capabilities of OVS models when data is collected sequentially. Code is available at: https://github.com/dongjunhwang/ConOVS

## 1  Introduction

In fields such as robotics [9] and autonomous driving [24, 47], there is a growing demand for models that can segment novel objects not included in the training dataset. However, conventional closed-set segmentation models, which are restricted to recognizing only the classes seen during training, fall short in meeting this demand. To address this limitation, Open-Vocabulary Segmentation (OVS) has emerged, aiming to enable segmentation of unseen classes that are not included in the training dataset. OVS continues to be an active area of research, particularly through methods that leverage foundation models such as CLIP [52, 58].

Most previous studies [49, 52, 55, 59] on OVS assume a scenario in which the model is trained once using a pre-training dataset. However, in practice, trainable datasets often arrive sequentially as new data are collected over time. Considering this setting, we first discuss how existing OVS models perform under such conditions. To facilitate a clearer discussion, we measure the relative performance of OVS models on seen and unseen classes using a *reference baseline*. We adopt OneFormer [18] for this role. It represents the state-of-the-art in closed-set segmentation and shares the same ConvNeXt backbone [27] as the OVS model [52], enabling a fair comparison in model capacity.

The most straightforward approach is to use the existing OVS model as is. In our experiments (Figure 1a), the existing OVS model achieves 86.4% of OneFormer's performance on the pre-training dataset COCO [26]. In contrast, its performance drops to 46.9% on a new dataset ADE20K [57], which contains unseen classes as well. These results indicate that the OVS model fails to perform well on datasets it has not encountered during training.

---

† Corresponding author (jschoe@sogang.ac.kr).

39th Conference on Neural Information Processing Systems (NeurIPS 2025).

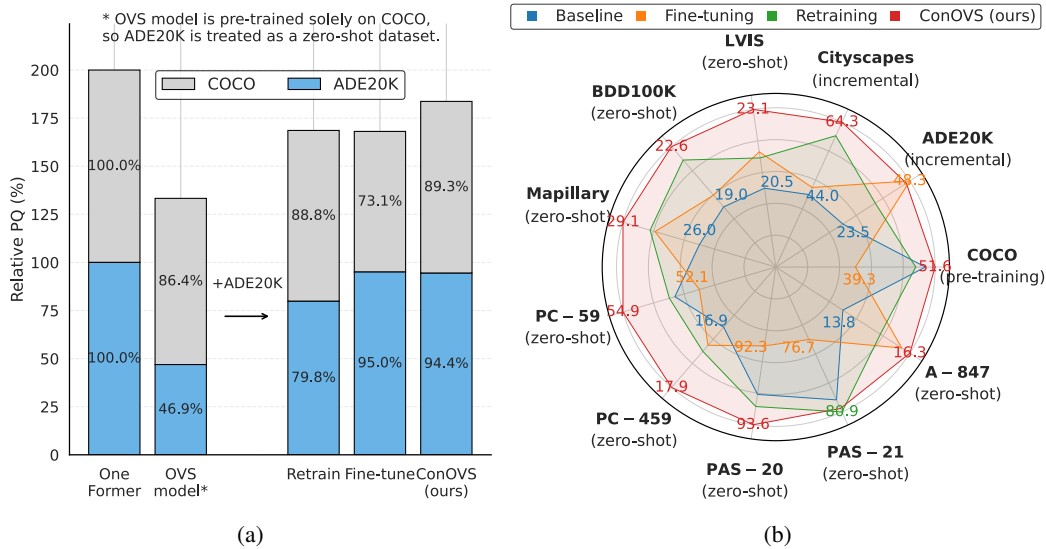

Figure 1: (a) Comparison of the performance of the OVS model (fc-clip [52]), Retraining, Fine-tuning, and ConOVS against the closed-set segmentation model OneFormer. (b) Performance of the Baseline (fc-clip [52]), Fine-tuning, Retraining, and ConOVS on the pre-training, incremental, and zero-shot test datasets. `PQ` is used.

To determine whether this performance gap is due to the inherent difficulty of the unseen classes or simply because the model has not been trained on them, we retrain the OVS model using both the pre-training dataset and the new dataset. As shown in Figure 1a, the model's performance on ADE20K improves significantly from 46.9% to 79.8% relative to OneFormer. This result confirms that the low performance on the new dataset is primarily due to the lack of exposure during training. It also suggests that this limitation of the OVS model can be effectively mitigated by training on newly collected data.

However, retraining the model from scratch demands substantial computational resources. In particular, this approach becomes impractical when the pre-training data is no longer accessible or computational resources are limited. To address these limitations, we consider an alternative approach: transfer learning. Specifically, we fine-tune the pre-trained OVS model on a new dataset. However, as shown in Figure 1a and 1b, this approach also has a limitation. It leads to performance degradation not only on the pre-training dataset but also on zero-shot tasks. This issue appears to stem from a well-known drawback of fine-tuning, namely, *catastrophic forgetting*. Therefore, we consider continual learning (CL) methods, which are designed to address catastrophic forgetting. However, most CL approaches are developed under the assumption that the number of classes is finite, making them unsuitable for open-vocabulary tasks where the number of classes can be potentially infinite [60, 61].

As a result, in scenarios where new datasets are continuously collected—as assumed in this paper—it is still unclear how to effectively utilize the incoming data, and finding a viable solution in OVS remains a non-trivial challenge. To address this, we propose ConOVS, a Mixture-of-Experts (MoE) based continual learning method that incrementally trains an existing OVS model on new datasets. Our method begins by fine-tuning the pre-trained OVS model to build a distinct expert for each new dataset. During inference, we estimate the probability that a given input sample is close to the distribution of each training dataset, based on their statistical representations. The model then computes an interpolation factor from these probabilities and dynamically combines the experts by interpolating their weights. This allows our method to produce an optimal model for predicting each input sample.

To simulate the scenario assumed in this paper, we sequentially introduce incremental datasets to an existing OVS model and evaluate the resulting models on three validation sets: the pre-training dataset, the incremental dataset, and zero-shot datasets. As shown in Figure 1b, our method not only significantly improves performance on the incremental dataset compared to standard retraining and fine-tuning, but also consistently enhances performance on both the pre-training and zero-shot datasets. Furthermore, compared to existing continual learning methods, our approach achieves superior performance across all three evaluation settings.

## 2    Related Works

### 2.1    Open-Vocabulary Segmentation

Recent open-vocabulary segmentation (OVS) research has focused on leveraging models capable of open-vocabulary classification, such as CLIP [36], to recognize classes that are not included in the training dataset. For example, fc-clip [52] identifies unseen classes by combining class embeddings from the model's decoder with those from CLIP. Moreover, a recent study [35] has explored an approach that retrieves LoRA modules trained on different datasets according to the input and utilizes them in conjunction with CLIP. Other methods further enhance the recognition of unseen classes by either applying visual grounding techniques like GradCAM [40] to CLIP [29, 42, 58] or distilling knowledge from both CLIP and the segmentation foundation model Segment Anything Model (SAM) [43, 54]. Meanwhile, there are also OVS approaches that do not rely on CLIP. For instance, methods such as X-Decoder [55, 62, 63] train both the encoder and decoder from scratch using segmentation datasets along with large-scale image–text pair datasets.

Most existing OVS studies are based on a scenario in which the model is trained only once. However, this setting inherently limits performance on unseen classes (see Section 1). To overcome this limitation, we analyze strategies for training OVS models in a scenario where new datasets are introduced sequentially.

### 2.2    Continual Learning

Acquiring additional knowledge in an already trained model is not straightforward. When a model is further trained on new data, it often tends to forget previously learned information while learning the new content [30]. This phenomenon is widely known as *catastrophic forgetting*. To address this issue, the field of continual learning (CL) has emerged. CL explores methods that enable models to learn from new data while retaining prior knowledge.

CL techniques are typically categorized into three types. First, replay-based methods store a subset of previously seen data and retrain the model using it to preserve prior knowledge [2, 38]. Second, regularization-based methods introduce penalty terms in the loss function to constrain parameter updates, preventing significant deviations during training on a new dataset [1, 21, 25]. Third, parameter-isolation-based methods mitigate interference by freezing previously learned parameters and allocating separate parameters for learning new data [20, 46]. Several approaches extend this idea into a Mixture-of-Experts (MoE) framework, where additional parameter sets are treated as distinct experts, and a gating module selects the appropriate expert based on the input [22, 45].

However, existing CL methods are designed under the assumption that the number of classes is finite, which limits their applicability in open-vocabulary settings [56, 60, 61]. Therefore, there is a need for novel approaches that enable continual learning in Open-Vocabulary Segmentation (OVS) scenarios, where new data are introduced incrementally. To address this, we propose a novel MoE-based continual learning technique that effectively expands the capacity of OVS models.

## 3    Motivation

In this section, we expand on the discussion from Section 1 and explore in greater detail how newly collected datasets can be leveraged to improve the performance of OVS models.

The most straightforward approach is to retrain the model from scratch using a joint dataset that combines the original and newly collected data. In practice, this strategy effectively preserves performance on seen classes while substantially improving performance on unseen classes. However, it suffers from two major limitations: (1) it incurs significant computational costs, as the model must be retrained from scratch every time new data are added; and (2) retraining becomes entirely infeasible if access to the original dataset has expired.

Due to these limitations, fine-tuning the model using only the newly collected dataset may appear to be a practical alternative. However, this approach compromises the model's original performance. As shown in Figure 2a, fine-tuning the OVS model results in a significant drop in performance not only on the pre-training dataset but also on the zero-shot test dataset. Qualitative examples provided in the

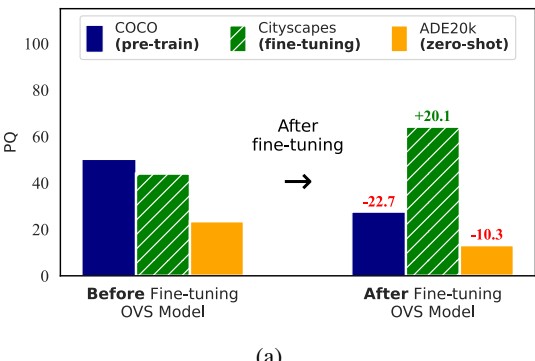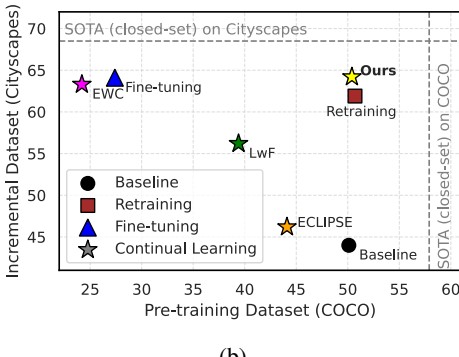

(a)                                         (b)

Figure 2: (a) Performance degradation on the pre-training and zero-shot datasets after fine-tuning. fc-clip is used. (b) Comparison of the performance of OneFormer [18], the baseline (fc-clip [52]), retraining, fine-tuning, three existing continual learning methods [20, 21, 25], and ConOVS on the pre-training and incremental datasets. All methods use the same iterations. PQ is used.

Appendix I further illustrate this phenomenon. This degradation is likely caused by a well-known issue in fine-tuning, known as catastrophic forgetting [21, 25].

Another potential direction is to apply continual learning (CL) methods to OVS models. However, most existing CL methods are built on the assumption of a finite set of classes, making them difficult to directly apply to open-vocabulary tasks [56, 60, 61]. For instance, [1, 20] apply CL to segmentation tasks by treating all unseen classes as background, which fundamentally conflicts with the goal of OVS models that aim to recognize potentially unlimited categories.

Even when existing CL methods are adapted for OVS (see Appendix A.2 for implementation details), our experimental results show that their effectiveness is limited. As shown in Figure 2b, OVS models trained with adapted CL methods perform significantly worse than the closed-set segmentation model OneFormer on both the pre-training and incremental datasets. We believe this arises because existing CL methods assume a closed-set segmentation with a finite label space, whereas OVS involves a potentially infinite label space, which these methods do not account for.

To address these issues, we propose **ConOVS**, a new continual learning method that sequentially improves the performance of OVS models. Specifically, ConOVS (1) reduces training cost by using only newly collected data, unlike retraining; (2) avoids catastrophic forgetting, unlike fine-tuning; and (3) effectively improves performance on the incremental and zero-shot test dataset, unlike existing CL methods.

## 4 Background

**Open-Vocabulary Segmentation (OVS)** aims to predict segmentation mask–class pairs from an input image $x_{img}$ and a text description $x_{text}$, which may include both seen (trained) and unseen classes. OVS models typically consist of three components: an image encoder, a text encoder, and a decoder, denoted as $f = \{f_{img}, f_{text}, f_{dec}\}$. The image encoder $f_{img}$ produces an image embedding $z_{img}$, and the text encoder $f_{text}$ produces a text embedding $z_{text}$. These are fed into the decoder $f_{dec}$, which, given $N$ learnable object queries, outputs $N$ pairs of predicted masks and class embeddings, $\{(m_i, c_i)\}_{i=1}^{N}$. Each $m_i$ is a predicted mask, and $c_i$ is its associated class embedding. Final class labels are assigned by matching each $c_i$ to the most similar text embedding.

**Continual Learning Setup.** We consider a continual learning scenario in which datasets containing new classes arrive sequentially, and the set of seen classes gradually expands over time. The model $f$ is first trained on a pre-training dataset $\mathcal{D}_{pre}$, and then incrementally updated using a sequence of datasets $\mathcal{D}_{inc,1}, \mathcal{D}_{inc,2}, \cdots$. At each time step $t \in \{1, 2, \cdots, n\}$, the model is trained only on $\mathcal{D}_{inc,t}$, without access to $\mathcal{D}_{pre}, \mathcal{D}_{inc,1}, \cdots, \mathcal{D}_{inc,t-1}$. The class set $\mathcal{C}_t$ from each incremental dataset is added to the previously seen class set, resulting in $\mathcal{C}_{seen} = \bigcup_{s=1}^{t} \mathcal{C}_s \cup \mathcal{C}_{pre}$. The model is evaluated on the test sets of all datasets up to time $t$ to assess both its ability to learn new classes and retain prior knowledge. To additionally evaluate generalization, we use a zero-shot test set $\mathcal{D}_{zero}$ containing unseen classes $\mathcal{C}_{unseen} \subset \mathcal{C}_{total} \setminus \mathcal{C}_{seen}$ that never appeared during training.

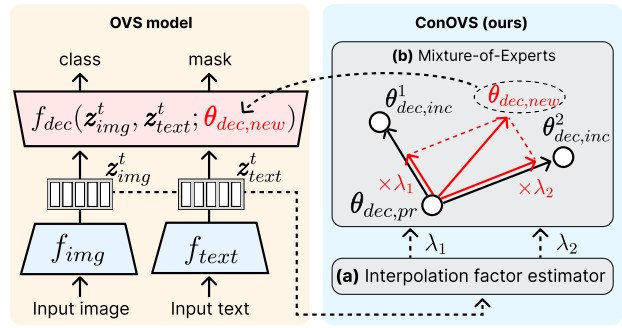

Figure 3: Overview of the inference process of our proposed method.

**Algorithm 1** Interpolation factor estimator

**Require:** Input $(\boldsymbol{x}_{\text{img}}, \boldsymbol{x}_{\text{text}})$, encoders $f_{\text{img}}, f_{\text{text}}$, decoder $f_{\text{dec}}$; MVN parameters $\{\boldsymbol{\Phi}^i_{\text{img}}, \boldsymbol{\Phi}^i_{\text{text}}\}^n_{i=0}$; PDF $p(\cdot|\boldsymbol{\Phi})$
**Ensure:** Interpolation factor $\boldsymbol{\lambda}$
1: Extract embeddings: $\boldsymbol{z}_{\text{img}} \leftarrow f_{\text{img}}(\boldsymbol{x}_{\text{img}})$, $\boldsymbol{z}_{\text{text}} \leftarrow f_{\text{text}}(\boldsymbol{x}_{\text{text}})$
2: Estimate likelihoods: $\boldsymbol{l}_{\text{img}} \leftarrow \{p(\boldsymbol{z}_{\text{img}} \mid \boldsymbol{\Phi}^i_{\text{img}})\}$, $\boldsymbol{l}_{\text{text}} \leftarrow \{p(\boldsymbol{z}_{\text{text}} \mid \boldsymbol{\Phi}^i_{\text{text}})\}$
3: Compute: $\boldsymbol{p}_{\text{img}} \leftarrow \text{softmax}(\boldsymbol{l}_{\text{img}})$, $\boldsymbol{p}_{\text{text}} \leftarrow \text{softmax}(\boldsymbol{l}_{\text{text}})$
4: Combine: $\boldsymbol{\lambda} \leftarrow \max(\boldsymbol{p}_{\text{img}}, \boldsymbol{p}_{\text{text}})$
5: **return** $\boldsymbol{\lambda}$

## 5 The Proposed Method: ConOVS

In this section, we propose **ConOVS**, a novel MoE-based continual learning method designed to train OVS models in scenarios where new datasets are sequentially collected. For clarity, we describe the proposed method in two parts: *Training Phase* and *Inference Phase*.

### 5.1 Training Phase

During training, we derive *expert models* and *multivariate normal* (MVN) *distributions* for each dataset. Specifically, we first train an OVS model from scratch using the pre-training dataset. Then, we fine-tune only the decoder on each incremental dataset to obtain an expert model specific to that dataset. For each dataset, we also compute the mean and covariance matrix of the image and text embeddings, which define the MVN distributions. These are represented as $\boldsymbol{\Phi}^i_{\text{img}} = (\boldsymbol{\mu}^i_{\text{img}}, \boldsymbol{\Sigma}^i_{\text{img}})$ and $\boldsymbol{\Phi}^i_{\text{text}} = (\boldsymbol{\mu}^i_{\text{text}}, \boldsymbol{\Sigma}^i_{\text{text}})$ for each dataset $i \in \{0, \cdots, n\}$. Here, $i = 0$ corresponds to the pre-training dataset, while $i \in \{1, \cdots, n\}$ refers to the incremental datasets.

### 5.2 Inference Phase

We perform inference by dynamically combining expert models based on the MVN distributions derived during training. Specifically, we first compute task vectors $\boldsymbol{v}_i$ for each expert model, defined as the arithmetic difference between the decoder weights of the $i$-th incremental expert $\boldsymbol{\theta}^i_{\text{dec,inc}}$ and the pre-trained decoder weights $\boldsymbol{\theta}_{\text{dec,pr}}$. Given an input sample, we feed the image $\boldsymbol{x}_{\text{img}}$ and class descriptions $\boldsymbol{x}_{\text{text}}$ into the image and text encoders, respectively, to obtain the corresponding embeddings $\boldsymbol{z}_{\text{img}}$ and $\boldsymbol{z}_{\text{text}}$. We then evaluate the likelihoods of these embeddings under the MVN distributions for all datasets, and collect them into the vectors $\boldsymbol{l}_{\text{img}}, \boldsymbol{l}_{\text{text}} \in \mathbb{R}^{n+1}$.

After that, we apply the softmax operation to the log-likelihood vector to normalize the proximity scores of each domain into the $[0, 1]$ range. This decision is motivated by a prior study [16], which reported that merging performance degrades when the interpolation factor exceeds 1. Finally, we compute the element-wise maximum of the two probability vectors to obtain the final interpolation factor vector $\boldsymbol{\lambda} \in \mathbb{R}^{n+1}$. The detailed procedure is provided in Algorithm 1, and ablation studies on the choice of softmax and element-wise maximum are presented in Appendix F.

The final decoder weights $\boldsymbol{\theta}_{\text{dec,new}}$ are computed as:

$$\boldsymbol{\theta}_{\text{dec,new}} = \boldsymbol{\theta}_{\text{dec,pr}} + \sum_{i=1}^{n} \lambda_i \boldsymbol{v}_i. \tag{1}$$

That is, the decoder is dynamically constructed by linearly combining task vectors $\boldsymbol{v}_i$ with interpolation weights $\lambda_i$, relative to the pre-trained decoder (see Figure 3b). Note that while $\lambda_0$ is not directly used in this computation, it is included in the softmax operation and thus indirectly affects the other $\boldsymbol{\lambda}$ elements. As a result, when the input is close to the pre-training distribution, $\lambda_0$ approaches 1, pushing the remaining $\lambda_i$ values toward 0.

The effectiveness and justification of this design are empirically validated in Section 6.

Table 1: Comparison of performance across Baselines (fc-clip, X-Decoder), Retraining, Fine-tuning, four existing continual learning methods, and ConOVS when the incremental dataset is (a) Cityscapes or (b) ADE20K. `PQ` is used.

| | | (a) Cityscapes | | | | | (b) ADE20K | | |
| --- | --- | --- | --- | --- | --- | --- | --- | --- | --- |
| Method | CL | COCO (pre-training) | Cityscapes (incremental) | ADE20K (zero-shot) | Method | CL | COCO (pre-training) | ADE20K (incremental) | Cityscapes (zero-shot) |
| fc-clip | ✗ | 50.1 | 44.0 | 23.5 | fc-clip | ✗ | 50.1 | 23.5 | 44.0 |
| Fine-tuning | ✗ | -22.7 | +20.1 | -10.3 | Fine-tuning | ✗ | -7.7 | **+24.1** | -3.0 |
| Retraining | ✗ | **+0.6** | +17.9 | +1.7 | Retraining | ✗ | +1.4 | +16.5 | -1.2 |
| ER | ✓ | -1.6 | +19.0 | +0.3 | ER | ✓ | +0.4 | +21.5 | -3.5 |
| LwF | ✓ | -10.7 | +12.2 | -0.8 | LwF | ✓ | -3.8 | +13.7 | -1.0 |
| EWC | ✓ | -25.9 | +19.3 | -9.8 | EWC | ✓ | -11.1 | +20.7 | -2.6 |
| ECLIPSE | ✓ | -6.0 | +2.2 | +0.9 | ECLIPSE | ✓ | -0.5 | +0.2 | -5.9 |
| **ConOVS (ours)** | ✓ | +0.3 | **+20.2** | **+2.5** | **ConOVS (ours)** | ✓ | **+1.7** | +23.8 | **+0.9** |
| X-Decoder | ✗ | 56.7 | 36.3 | 16.7 | X-Decoder | ✗ | 56.7 | 16.7 | 36.3 |
| Fine-tuning | ✗ | -50.4 | **+26.6** | -12.9 | Fine-tuning | ✗ | -37.3 | +28.2 | -3.7 |
| **ConOVS (ours)** | ✓ | **-0.4** | **+26.6** | **+0.1** | **ConOVS (ours)** | ✓ | **-1.5** | **+29.2** | **+1.4** |

## 6 Experiments

**Learning Sequences.** This study assumes a scenario where trainable datasets arrive sequentially and evaluates OVS models that are incrementally trained on them. In the main paper, we examine three learning sequences. In Scenario 1 (**S1**), the model is pre-trained on COCO [26], incrementally trained on Cityscapes [7], and evaluated on ADE20K [57] as the zero-shot test set. In Scenario 2 (**S2**), the model is again pre-trained on COCO but incrementally trained on ADE20K, with Cityscapes used for zero-shot evaluation. In Scenario 3 (**S3**), the model is pre-trained on COCO and incrementally trained on both Cityscapes and ADE20K. For zero-shot evaluation, we use a diverse collection of datasets: LVIS [10], BDD100K [51], Mapillary Vistas [33], PC-59, PC-459 [31], PAS-20, PAS-21 [8], and A-847 [57]. We further validate our method on a larger number of incremental datasets in Scenario 4 (**S4**), with the results provided in Appendix E. Evaluation is conducted on the test sets of the pre-training and incremental datasets, as well as the designated zero-shot test sets.

**Implementation Details.** We apply our method to two OVS models: fc-clip with ConvNeXt-L [27] and X-Decoder with Focal-L [50]. During the pre-training phase, fc-clip trains only the decoder, while X-Decoder trains both the encoder and decoder. In the fine-tuning phase, both models train only the decoder. The temperature $T$ in the softmax is set to 0.01, and log-likelihood is used to compute probabilities from the MVN distributions. All experiments are run on two NVIDIA A5000 GPUs.

**Evaluation Metrics.** We evaluate panoptic, instance, and semantic segmentation using PQ, mAP, and mIoU, respectively. Due to space constraints, we report only PQ in the main paper, with the others in the Appendix J. Some zero-shot test datasets support only specific segmentation tasks; for example, LVIS supports only instance segmentation. In such cases, we evaluate performance only on the supported task.

### 6.1 Main Results

In this section, we compare the performance of the proposed ConOVS and other approaches under the three scenarios. We first analyze the results for scenarios S1 and S2, followed by scenario S3. We then provide a more in-depth analysis of our method, including an investigation into the behavior of the interpolation factors. All methods were trained with the same number of iterations to ensure a fair comparison, and detailed information on the training cost of each method is provided in Appendix D.1.

In scenarios **S1** and **S2**, where only a single incremental dataset is used for training, our method consistently outperforms existing approaches across all datasets, whether the incremental dataset is ADE20K or Cityscapes (see Table 1). In particular, compared to retraining, our method almost maintains or even improves performance on the pre-training dataset, despite not using it during additional training (e.g., Retraining: +1.4 vs. Ours: +1.7 in S2). It also achieves superior performance on the incremental dataset itself (e.g., Retraining: +16.5 vs. Ours: +23.8 in S2). Moreover,

compared to fine-tuning and conventional continual learning, our method improves performance on the incremental dataset without compromising performance on the pre-training dataset. This improvement is attributed to the dynamic interpolation of expert models in our method, which helps mitigate catastrophic forgetting.

Our method also achieves the best performance on the zero-shot test dataset. For instance, in scenario S2, performance on the Cityscapes improves by +0.9, whereas all other methods show performance drops. This result indicates that our method enhances recognition of a wider range of classes while preserving previously learned knowledge.

In scenario **S3**, our method consistently achieves superior performance compared to both fine-tuning and retraining. Specifically, as shown in Table 2, fine-tuning performs well only on the most recently trained dataset, whereas our method consistently achieves strong results on all three datasets. By contrast, retraining shows lower performance than our method, likely due to its need for more iterations to converge. In comparison, our method yields better results with

Table 2: Performance comparison in scenario S3. The best performance for each dataset is underlined. "City→ADE" means fine-tuning on Cityscapes first, then ADE20K. `PQ` is used.

| Method | Learning Sequence | COCO (pre-training) | ADE20K (incremental) | Cityscapes (incremental) |
|---|---|---|---|---|
| fc-clip | - | 50.1 | 23.5 | 44.0 |
| Fine-tuning | ADE → City | 20.8 | 15.4 | 65.2 |
| Fine-tuning | City → ADE | 39.3 | 48.3 | 46.0 |
| Retraining | COCO, City, ADE | 48.6 | 35.5 | 60.5 |
| **ConOVS (ours)** | City, ADE | **51.6** | **47.0** | **64.3** |

the same number of training iterations, demonstrating greater training efficiency. Note that the analysis related to the number of training iterations in retraining is provided in Appendix G.3.

Table 3: Performance comparison on 8 unseen datasets in scenario S3. The best performance for each dataset is underlined. `PQ` is used.

| Method | Learning Sequence | LVIS (mAP) | BDD100K (PQ) | Mapillary (mIoU) | PC-59 (mIoU) | PC-459 (mIoU) | PAS-20 (mIoU) | PAS-21 (mIoU) | A-847 (mIoU) |
|---|---|---|---|---|---|---|---|---|---|
| fc-clip | - | 20.5 | 19.0 | 26.0 | 53.0 | 16.9 | 93.1 | 80.2 | 13.8 |
| Fine-tuning | City → ADE | 21.7 | 19.7 | 27.8 | 52.1 | 17.2 | 92.3 | 76.7 | 16.0 |
| Fine-tuning | ADE → City | 10.4 | 21.3 | 24.2 | 45.9 | 13.5 | 87.4 | 70.7 | 11.5 |
| Retraining | COCO, City, ADE | 21.5 | 21.8 | 28.0 | 53.2 | 17.3 | 93.3 | 80.9 | 15.2 |
| **ConOVS (ours)** | City, ADE | **23.1** | **22.6** | **29.1** | **54.9** | **17.9** | **93.6** | **80.7** | **16.3** |

In addition, our method also consistently outperforms other approaches in various zero-shot evaluations. As shown in Table 3, it achieves superior performance across all eight zero-shot test datasets. This result suggests that the dynamic interpolation of expert models in our method facilitates recognition of a broader range of unseen classes.

Table 4: Comparison of performance on seen and unseen classes in the zero-shot test dataset ADE20K. `mIoU` is used. (b) Comparison of PQ, SQ, and RQ between fc-clip and ConOVS in the zero-shot test dataset ADE20K.

<table>
<tr><td colspan="3" align="center">(a)</td><td colspan="4" align="center">(b)</td></tr>
<tr><th>Method</th><th>Seen Classes</th><th>Unseen Classes</th><th>Method</th><th>PQ</th><th>SQ</th><th>RQ</th></tr>
<tr><td>fc-clip</td><td>35.0 (+0.0)</td><td>28.6 (+0.0)</td><td>fc-clip</td><td>23.5 (+0.0)</td><td>61.7 (+0.0)</td><td>28.3 (+0.0)</td></tr>
<tr><td>ConOVS (ours)</td><td>37.9 (+2.9)</td><td>30.9 (+2.3)</td><td>ConOVS (ours)</td><td>25.9 (+2.4)</td><td>73.1 (+11.4)</td><td>31.2 (+2.9)</td></tr>
</table>

**Evaluation of the Truly Unseen Classes.** Some classes in the zero-shot test datasets may overlap with those in the training data. For instance, ADE20K shares 38 of its 150 classes with COCO. To more accurately assess zero-shot performance, we separately evaluate the model on truly unseen classes that do not appear in the training data. Therefore, we split ADE20K into seen and unseen subsets and measure performance on each in scenario S1.

As shown in Table 4a, our method improves performance by a similar margin on both seen and unseen classes (seen: +2.9, unseen: +2.3). This suggests that the performance gain is not solely from improved recognition of seen classes, but also reflects better generalization to unseen classes.

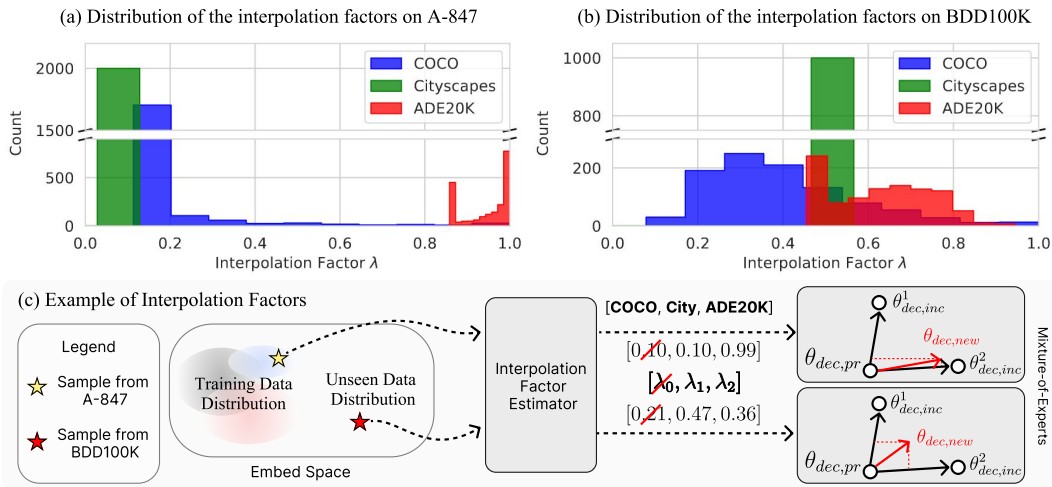

Figure 4: Interpolation factor behavior across different input sample distributions.

**Analysis of Improvements in Unseen Classes.** To better understand the source of performance improvements in unseen classes, we analyzed results on the zero-shot dataset ADE20K by comparing the PQ, SQ, and RQ scores of the baseline and our proposed method. As shown in Table 4b, incorporating ConOVS into the baseline model improves both PQ and RQ. The most notable gain, however, is observed in SQ, which evaluates the quality of the predicted segmentation masks. These results indicate that the improvements in unseen classes are primarily driven by enhanced segmentation quality rather than improved mask classification.

**Understanding the Behavior of the Interpolation Factor.** We analyze how the proposed method adapts to different input sample distributions. To this end, we examine the distribution of interpolation factors $\lambda$ estimated by the interpolation factor estimator across two zero-shot test datasets. One is A-847, which shares a similar distribution with the incremental training dataset ADE20K, and the other is BDD100K, which differs significantly from all training datasets.

As shown in Figure 4a, the interpolation factors for A-847 tend to be close to 0 or 1. In particular, the expert trained on ADE20K receives a $\lambda$ value close to 1, while other experts receive values close to 0. This shows that when input samples are similar to a previously trained distribution, our method selectively activates the corresponding expert to maximize performance (see Figure 4c top-right).

In contrast, as illustrated in Figure 4b, the interpolation factors for BDD100K are more evenly distributed between 0 and 1. This suggests that the input samples do not clearly belong to any of the known training distributions. In such cases, our method disperses the $\lambda$ values to avoid over-reliance on a single expert. Instead, it combines the weights of multiple experts based on the probability that the input sample belongs to each distribution. This allows the model to leverage knowledge from various datasets and produce more accurate predictions even for samples from unfamiliar domains (see Figure 4c bottom-right).

## 6.2 Ablation Study

In this section, we conduct ablation studies to analyze the contribution of each component in the proposed method. All experiments are conducted in scenario S1.

**Ablation Study of Image and Text Distribution.** Our method computes the interpolation factor of an input sample using the MVN distributions of image and text embeddings for each training dataset. To analyze how the interpolation factors are affected by the distribution design, we compare three configurations: image only, text only, and combined image-text.

As shown in Table 5a, using both image and text distributions yields the best performance on the incremental dataset. This suggests that combining both modalities enables more accurate estimation of the input sample's proximity to training distributions, leading to better expert selection.

Table 5: (a) Comparison of the interpolation factor estimator when using both image and text distributions versus using only one of them. `PQ` is used. (b) Performance comparison when the MVN distribution is replaced with K-means clustering or KDE. fc-clip and `PQ` are used.

(a)

| Distribution | COCO (pre-training) | Cityscapes (incremental) | ADE20K (zero-shot) |
|---|---|---|---|
| image only | 51.5 | 43.4 | 25.8 |
| text only | **51.9** | 60.7 | 25.9 |
| image + text | 51.6 | **64.3** | 26.0 |

(b)

| Methods | COCO (pre-training) | Cityscapes (incremental) | ADE20K (zero-shot) |
|---|---|---|---|
| k-means clustering | 42.4 | 64.1 | 26.1 |
| kernel density estimation | 48.1 | 57.4 | 26.1 |
| MVN distribution | **50.4** | **64.3** | 26.0 |

**Evaluating Alternative Approaches against the MVN Distribution.** We evaluate and compare two alternative techniques to the MVN distribution used in our method for estimating interpolation factors. Specifically, we replace the MVN distribution with K-means clustering or Kernel Density Estimation (KDE), and analyze the resulting performance changes. Detailed descriptions of the K-means and KDE are provided in the Appendix B.5.

As shown in Table 5b, both K-means and KDE yield lower performance on the pre-training and incremental dataset. These results suggest that the MVN distribution enables more accurate estimation of interpolation factors for in-distribution data. We attribute this to its relatively simple structure and low dimensionality, which make it less sensitive to outliers than K-means or KDE.

**Replacing Softmax with Argmax.** The proposed method uses the softmax function to compute interpolation factors for each dataset. We compare the performance on eight zero-shot datasets when replacing the softmax function with the argmax operation. Table 6 presents the evaluation results. The experimental results show that softmax consistently outperforms argmax across all zero-shot datasets (e.g., on LVIS, argmax: 21.3, softmax: 23.1).

Table 6: Performance comparison between the argmax and softmax operations in the interpolation factor estimator. We use fc-clip with our method and fine-tune it on both Cityscapes and ADE20K. `PQ` is used.

| Decision Rule | Incremental Dataset | LVIS (mAP) | BDD100K (PQ) | Mapillary (mIoU) | PC-59 (mIoU) | PC-459 (mIoU) | PAS-20 (mIoU) | PAS-21 (mIoU) | A-847 (mIoU) |
|---|---|---|---|---|---|---|---|---|---|
| Argmax | Cityscapes, ADE20k | 21.3 | 18.3 | 26.9 | 53.1 | 17.0 | 93.2 | 80.2 | **16.3** |
| Softmax | Cityscapes, ADE20k | **23.1** | **22.6** | **29.1** | **54.9** | **17.9** | **93.6** | **80.7** | **16.3** |

Specifically, on datasets such as LVIS and BDD100K, softmax demonstrates clearly superior performance. However, for PAS-20, PAS-21, and A-847, the performance difference between softmax and argmax is minimal. This occurs because, when the input sample is close to the distribution of the pre-training or incremental dataset, the interpolation factor obtained from softmax tends to be close to 0 or 1. As a result, softmax behaves similarly to argmax.

**Hyperparameter Sensitivity Analysis.** Our method uses a softmax operation to compute the interpolation factor, and we analyze the effect of the softmax temperature hyperparameter $T$. The temperature $T$ directly influences the distribution of the interpolation factor: a low $T$ smooths the factor values, while a high $T$ pushes them toward extreme values of 0 or 1. Table 7 summarizes how this behavior affects performance.

When $T$ is small, the interpolation factor $\lambda$ becomes overly smoothed, which prevents the ex-

Table 7: Effect of softmax temperature $T$ on performance across datasets. `mIoU` is used.

| $T$ | COCO (pre-training) | ADE20K (incremental) | Cityscapes (zero-shot) | Total |
|---|---|---|---|---|
| 0.0001 | 50.7 | 35.4 | 43.8 | 129.9 |
| 0.001 | 51.2 | 42.2 | **43.9** | 137.3 |
| 0.01 | **51.8** | 47.3 | 43.7 | **142.8** |
| 0.1 | 51.3 | **47.5** | 43.2 | 142.0 |
| 1.0 | 51.2 | 47.4 | 43.2 | 141.8 |

pert models for each dataset from being utilized. This leads to performance degradation on the incremental dataset. In contrast, when $T$ is large, $\lambda$ converges to values close to 0 or 1, resulting in the selective use of a single expert model. This degrades performance on the zero-shot dataset. These findings suggest that appropriately integrating multiple models is essential for effective generalization to zero-shot datasets, and that extreme interpolation factors hinder this process.

**Decoder Interpolation.** Unlike our method, which fine-tunes the entire decoder for each dataset, existing MoE-based continual learning methods [22, 45] primarily adopt Visual Prompt Tuning (VPT), where only a small subset of parameters is trained for each incremental dataset. This approach differs from ours in two key aspects: expert models consist of only partial decoder parameters, and a single expert is selected at inference time instead of performing interpolation. To assess the effectiveness of our full decoder fine-tuning strategy, we replace it with the VPT-based approach and compare their performance.

Specifically, we implement the prompt tuning method based on [45] as follows: (1) for each incremental dataset, we train only the decoder's object queries and positional embeddings and store them in a prompt pool; (2) during inference, we compute interpolation factors for each dataset using the same procedure as our method; (3) we identify the dataset with the highest interpolation factor; and (4) retrieve the corresponding object queries and positional embeddings from the prompt pool and apply them to the decoder for prediction.

Table 8: Performance comparison when the decoder interpolation in our method is replaced with a visual prompt tuning-based approach. fc-clip and `PQ` are used.

| Method | COCO (pre-training) | Cityscapes (incremental) | ADE20K (zero-shot) |
|---|---|---|---|
| Prompt Tuning | 43.3 | 48.9 | 24.4 |
| Decoder Interpolation | **50.4** | **64.3** | **26.0** |

As shown in Table 8 and the experimental results, the prompt tuning variant consistently underperforms our method across pre-training, incremental, and zero-shot test datasets. This suggests that full decoder fine-tuning enables more effective adaptation to new datasets compared to VPT, which is constrained by its limited number of trainable parameters. Moreover, interpolating multiple experts provides greater flexibility and representational power than selecting a single expert, further supporting the advantage of our approach.

# 7 Limitation

Our method generates a unique decoder weight for each input sample, which can limit its applicability when the inference batch size exceeds one—a common constraint in other MoE-based continual learning approaches [41, 45]. However, since only the decoder varies per input and the encoder is shared across samples, the encoder can process inputs in batches. The resulting embeddings are then decoded individually using their corresponding weights. This design reduces the batch size limitation by supporting batched encoder processing and per-sample decoding.

# 8 Conclusion

This paper identifies the performance limitations of existing Open-Vocabulary Segmentation (OVS) methods on unseen data, an aspect that has been largely overlooked in prior work. To address this issue, we introduce a new learning scenario in which newly collected datasets are incrementally used to further train the OVS model. Under this setting, we show that conventional approaches—such as retraining, fine-tuning, and continual learning—are either impractical or difficult to apply effectively.

To overcome these challenges, we propose **ConOVS**, a novel MoE-based continual learning method for OVS. In ConOVS, predictions are made by dynamically combining the decoders of expert models based on the probability that the input sample belongs to the distribution of each training dataset. We validate the effectiveness of our method through extensive evaluations across various sequential learning scenarios and compare it against existing approaches. Experimental results show that ConOVS consistently achieves superior performance on pre-training, incremental, and zero-shot test datasets, demonstrating its ability to effectively expand the recognition capability of OVS models.

**Broader Impacts.** The proposed method can be applied to real-world applications such as robotics, where new objects continuously appear in the environment. However, if the pre-training dataset is biased, the model may continue to produce skewed predictions even after additional training, as it is explicitly designed to preserve previously learned knowledge. It is therefore important to be aware of this characteristic of the proposed technique, as a lack of such awareness may lead to unexpected model behavior.

## Acknowledgement

We would like to thank Yeji Park, Beomyun Kwon, and Joonkyung Kim for the insightful discussions and valuable feedback during the development of this work. This work was partly supported by the Institute of Information & Communications Technology Planning & Evaluation (IITP) grant funded by the Korea government(MSIT) (No. RS-2025-25441313, Professional AI Talent Development Program for Multimodal AI Agents, Contribution: 50%) and the National Research Foundation of Korea (NRF) grant funded by the Korea government (MSIT) (No. RS-2024-00350430, Mitigating Hallucinations for Trustworthy Large Vision-Language Model: Datasets, Evaluation, Learning, and Inference, Contribution: 50%).

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
