# OpenReview forum: "OVS Meets Continual Learning: Towards Sustainable Open-Vocabulary Segmentation"
_NeurIPS.cc/2025/Conference — NeurIPS 2025 poster_

### Official Review · Reviewer_aa5k · 2025-07-02

**Clarity:** 1
**Significance:** 2
**Originality:** 2
**Rating:** 4
**Confidence:** 2

**Summary:**

This paper aims to solve the problem of continual learning of Open-Vocabulary Semantic Segmentation (OVSS), a scenario where new datasets are added without access to previously used training data. To address this, the paper trains a mixture of experts, with each expert specifically trained on a distinct dataset. It then balances the influence of these experts by calculating the proximity between the embeddings of the image being inferred and the embeddings representing each dataset. For example, if an image's embedding is closer to a particular dataset, a higher weight is assigned to the expert trained on that dataset. The paper's experiments used Cityscapes and ADE20K as incremental datasets.

**Questions:**

Overall, while I am not well-versed in the literature of Open-Vocabulary Semantic Segmentation (OVSS) with continual learning and have low confidence in my decision, I believe the writing of Section 5.2 should be significantly improved, and more data should be tested for continual learning to convince the reader. Regarding SemLA, considering that the paper was released in March 2025, a discussion of it would be sufficient; it's not a significant reason for rejection. Overall, I am leaning towards rejection due to the experimental setup and writing clarity. As I'm unfamiliar with the topic, I will carefully listen to other reviewers' opinions and make a final decision.

**Ethical Concerns:**

["NO or VERY MINOR ethics concerns only"]

**Final Justification:**

The authors have provided solid evidence to demonstrate generalizability with more datasets. All of my concerns have been resolved.

**Limitations:**

yes

**Paper Formatting Concerns:**

No formatting concerns

**Quality:**

2

**Strengths And Weaknesses:**

**Strength**
1. Well-written introduction part.

---
**Major Weakness**
1. **Conceptual similarity to Qorbani et al., "Semantic Library Adaptation: LoRA Retrieval and Fusion for Open-Vocabulary Semantic Segmentation" (CVPR '25).**
Aside from the difference in the specific task, the conceptual similarity between this paper and SemLA is very high and warrants extensive discussion in the related works section. Both approaches calculate proximity to each training dataset to determine weights between datasets (similar to ConOVS). With these weights, they merge LoRA weights (analogous to Mixture-of-Experts in ConOVS).


2. **The writing in Section 5.2, "Inference Phase," is puzzling.** It would be much easier to understand if the shapes of the vectors were notated. Additionally, it is almost impossible for me to understand what is happening in this stage upon a first read. The paper states, "evaluate the likelihoods of these embeddings under the MVN distributions for all datasets, and collect them into the vectors," but the method for doing so is unclear. What are the shapes of $l_{img}$ and $l_{text}$? After several re-readings, I could infer that the shape for the vectors might be $\mathbb{R}^n$, with each element representing the likelihood of the image and text being close to the $i$-th dataset. Why is softmax conducted here? Why is a max operation performed? While I can infer the authors' intention, it takes a significant amount of time to reach that conclusion. Furthermore, if the purpose of using softmax is to create a probability distribution, then why do the authors perform an element-wise maximum that violates the assumption that their sum is 1? Overall, the writing in this section should be significantly improved for easier comprehension by readers.

3. **Regarding the experimental setup, the number of incremental datasets (n) used is too low to convincingly demonstrate the method's effectiveness.** For instance, in Table 1, the number of incremental datasets is only one, which makes it difficult to assess the method's generalizability to a larger n. In contrast, SemLA demonstrated effectiveness by training with 10 datasets and testing with 20 benchmarks. Given this, there seems to be ample opportunity to integrate more datasets into the evaluation.

---
**Minor Weakness**
- In Line 127, it would be beneficial if the "characteristics of OVS tasks" could be elaborated upon.

- For Figure 2(b), it would be helpful to include citations for competitors in the caption.

---

> ### Author Rebuttal · Authors · 2025-07-31
>
> Thank you for your detailed review and constructive feedback. We appreciate your comments on the writing, experimental setup, and relation to SemLA, and address each point below.
>
> ---
>
> ### **Q1) This paper and SemLA share a core idea: estimating dataset proximity to compute weights for merging LoRA modules, similar to ConOVS’s MoE approach.**
>
> SemLA \[1\] was released on arXiv in March 2025 and presented at CVPR in June 2025\. According to the NeurIPS Call for Papers, it is officially considered a contemporaneous work.
>
> Nonetheless, we agree with the reviewer that a discussion on the conceptual connection between SemLA and our method is necessary. As the reviewer pointed out, there is a conceptual similarity between the two approaches—both compute the proximity of an input image to each training dataset and use it to determine dynamic merging weights during inference. However, our contributions differ from those of SemLA in several important aspects:
>
> - Our paper is the first to identify and address the limitations of existing open-vocabulary segmentation (OVS) methods in continual learning (CL) scenarios, where new datasets are continuously collected over time. Highlighting the issues of OVS under CL is an important contribution of our work. To the best of our knowledge, this setting has not been considered in SemLA, which focuses on single-stage inference without addressing continual dataset shifts.
>
> - Compared to SemLA, our method adopts different design choices in three key aspects: (1) it uses both image and text embeddings to assess domain relevance, whereas SemLA relies solely on image embeddings; (2) it estimates domain proximity using multivariate normal (MVN) distributions, while SemLA computes L2 distances to dataset-specific centroids; and (3) it fine-tunes the decoder for each dataset, in contrast to SemLA, which applies LoRA modules to the CLIP image encoder. A detailed explanation of the first point is provided in our response to Q2 below.
>
> In addition, we implemented SemLA within our experimental setup for a direct comparison. Specifically, we modified our framework to follow SemLA’s key design choices: (1) computing proximity using only image embeddings, (2) estimating domain proximity by calculating the L2 distance between the input embedding and dataset-specific centroids, and (3) replacing decoder fine-tuning with LoRA fine-tuning on the CLIP image encoder.
>
> As shown in the table below, our method (ConOVS) outperforms this SemLA variant on the incremental dataset. However, since multiple design differences exist between the two methods, it is difficult to isolate the exact cause of the performance gap. A deeper analysis of this result was challenging to conduct within the limited rebuttal period, but we consider it an important direction for future work. Nonetheless, the observed improvement suggests that our design choices might be more suitable for continual open-vocabulary segmentation.
>
> | method | coco (pretraining) | cityscapes (incremental) | ade20k (zero-shot) |
> | :---- | :---: | :---: | :---: |
> | fc-clip | 50.1 | 44.0 | 23.5 |
> | SemLA | **50.9** | 61.1 | **26.0** |
> | ConOVS (ours) | 50.4 | **64.4** | **26.0** |
>
> We will add the discussion on the relation between SemLA and our method to Section 2 (*Related Works*), and the experimental results to Section 6 (*Experiments*). We appreciate the insightful feedback.
>
> > \[1\] Qorbani, Reza, et al. *Semantic Library Adaptation: LoRA Retrieval and Fusion for Open-Vocabulary Semantic Segmentation.* CVPR 2025\.
>
> ---
>
> ### **Q2) Section 5.2 lacks clarity in vector shapes and operations like softmax and max; the rationale should be explained more clearly.**
>
> We agree with the reviewer that the description in Section 5.2 was difficult to follow. In this response, we first clarify the rationale behind our use of softmax and element-wise maximum operations, and then provide details on the improvements we made to Section 5.2 for better readability.
>
> We apply the softmax operation to the log-likelihood vector to normalize the proximity scores of each domain to the \[0, 1\] range. This decision is based on a prior study \[2\], which found that when the interpolation factor exceeds 1, merging performance degrades. To normalize the log-likelihood values obtained from the MVN distribution, we considered three strategies, including min-max normalization, sigmoid, and softmax. As shown in the table below, softmax achieved the best performance, which led us to adopt it.
>
> | Operation | coco (pretraining) | cityscapes (incremental) | ade20k (zero-shot) | Average |
> | :---- | :---- | :---- | :---- | :---- |
> | minmax | 50.3 | 50.5 | **26.1** | 42.3 |
> | sigmoid | 50.3 | 50.3 | 26.0 | 42.2 |
> | softmax | **50.4** | **64.4** | 26.0 | **46.9** |
>
> We use the element-wise maximum operation to combine information from both the image and text modalities. Initially, we considered whether to rely on only the image domain, only the text domain, or both. For combining both modalities, we evaluated three options: average, multiplication, and element-wise maximum. As shown in the table below, the results show that combining both modalities performs better than using a single modality, and among the combination methods, element-wise maximum achieved the best performance. Based on these findings, we selected element-wise maximum as our fusion strategy.
>
> |  | Operation | coco (pretraining) | cityscapes (incremental) | ade20k (zero-shot) | Average |
> | :---- | :---- | :---- | :---- | :---- | :---- |
> | image only | \- | 51.5 | 43.4 | 25.8 | 40.2 |
> | text only | \- | **51.9** | 60.7 | 25.9 | 46.2 |
> | image and text | average | 50.3 | 64.2 | 25.9 | 46.8 |
> | image and text | multiplication | 50.1 | 63.8 | 25.8 | 46.6 |
> | image and text | maximum | 50.4 | **64.4** | **26.0** | **46.9** |
>
> We acknowledge that the use of element-wise maximum does not guarantee that the resulting proximity values sum to one. However, this aligns with prior practice in task vector-based merging methods \[2,3\], where interpolation factors are not required to sum to one. In our experiments, we observed no adverse effects resulting from this design choice.
>
> To improve the clarity of Section 5.2, we not only added the above explanations regarding our design choices but also provided more concrete details, such as the dimensions of the vectors. For example, we recognized that the definitions of the likelihood vectors $l\_{\\text{text}}$ and $l\_{\\text{img}}$ were ambiguous. We revised the text to clarify that these vectors represent the proximity values between the input sample and each dataset, and we explicitly stated their dimensions to improve readability.
>
> > \[2\] Ilharco, Gabriel, et al. *Editing models with task arithmetic.* ICLR 2023\.
> \[3\] Yadav, Prateek, et al. *Ties-merging: Resolving interference when merging models.* NeurIPS 2023\.
>
> ---
>
> ### **Q3) The number of incremental datasets is too small to convincingly demonstrate generalizability. More datasets should be included for a convincing evaluation.**
>
> We agree that evaluating our method with a larger number of incremental datasets is important for demonstrating generalizability. To address this, we designed a new scenario in which the model is sequentially trained on five datasets: COCO, Cityscapes, ADE20K, BDD100K, and Mapillary Vistas. While we aimed to include more datasets, our work focuses on panoptic segmentation, for which publicly available datasets are considerably more limited than those available for semantic segmentation as used in SemLA.
>
> The results of this extended experiment are shown in the table below. As can be seen, our method outperforms the baseline, fine-tuning, and retraining approaches. These results indicate that our method remains effective as the number of incremental datasets increases, demonstrating its scalability under complex domain shifts.
>
> |  | pre-training | incremental |  |  |  | zero-shot |  |  |  |  |  | Average |
> | :---- | :---: | :---: | :---: | :---: | :---: | :---: | :---: | :---: | :---: | :---: | :---: | :---: |
> |  | COCO | Cityscapes | A-150 | BDD100K | Mapillary | LVIS | PC-59 | PC-459 | PAS-20 | PAS-21 | A-847 |  |
> | fc-clip | 50.1 | 44.0 | 23.5 | 19.0 | 26.0 | 20.5 | 53.0 | 16.9 | 93.1 | 80.2 | 13.8 | 40.0 |
> | fine-tuning | 31.7 | 55.5 | 28.7 | 25.8 | 36.3 | 17.7 | 49.8 | 16.3 | 90.0 | 73.9 | 14.0 | 40.0 |
> | retraining | 49.0 | 62.3 | 36.3 | 28.2 | 34.5 | 21.7 | **53.6** | **17.4** | **94.0** | **80.7** | 15.6 | 44.8 |
> | ConOVS (Ours) | **50.1** | **62.9** | **38.5** | **29.1** | **35.0** | **21.8** | **53.6** | 17.3 | 93.2 | 80.4 | **16.0** | **45.3** |
>
> We will include this experiment and its discussion in Section 6.1 (*Main Results*). Thank you for the valuable feedback.
>
> ---
>
> ### **Q4) In Line 127, it would be beneficial if the "characteristics of OVS tasks" could be elaborated upon.**
>
> What we intended by "characteristics of OVS tasks" is the fundamental difference between open-vocabulary segmentation (OVS) and closed-set segmentation. Specifically, OVS tasks involve a potentially infinite label space, whereas closed-set segmentation assumes a fixed, finite label set. Taking this into account, we will revise the sentence as follows:
>
> > *We believe this arises because existing CL methods assume a closed-set segmentation with a finite label space, whereas OVS involves a potentially infinite label space, which these methods do not account for.*
>
> ---
>
> ### **Q5) For Figure 2(b), it would be helpful to include citations for competitors in the caption.**
>
> We agree with the reviewer and will update the caption of Figure 2(b) to include citations for the continual learning methods used. The revised caption is as follows:
>
> > *Figure 2: … (b) Comparison of the performance of OneFormer \[12\], the baseline (fc-clip \[44\]), retraining, fine-tuning, three existing continual learning methods \[14,15,18\], and ConOVS on the pre-training and incremental datasets.*

---

> > ### Comment · Reviewer_aa5k · 2025-08-05
> >
> > I greatly appreciate the authors' comprehensive rebuttal. All of my concerns have been resolved, and I will raise my score accordingly.

---

### Official Review · Reviewer_5TFJ · 2025-07-02

**Clarity:** 3
**Significance:** 3
**Originality:** 3
**Rating:** 4
**Confidence:** 4

**Summary:**

This paper presents ConOVS, a MoE-based framework for continual learning in the field of open-vocabulary segmentation. The paper first suggests baseline methods, being fine-tuning and retraining, to incorporate additional training data, which suffer from catastrophic forgetting and declines in performance for its original pre-trained dataset. In this regard, the authors propose ConOVS, which dynamically mixes the weight for the segmentation decoder during inference based on the encoder representation statistics. The authors demonstrate clear advantages compared to baselines, while also providing various experiments settings and ablations to verify the design choice.

**Questions:**

**[Q1]** Could the authors provide additional explanation or analysis on how the unseen classes are improving?
- To the reviewers perspective, one possible explanation might be that the fine-tuned models on the incremental datasets might have improved in identifying segmentation masks, as it would be hard to expect the mask classification to have improved. One possible approach to examine this could be inspecting the RQ and SQ for the panoptic datasets, and observe which is improving or not.

**[Q2]** Could the authors ablate the choice for only fine-tuning the decoder? Furthermore, could the proposed method be applicable to cost-based OVS methods that fine-tune the encoder[1,2,3], e.g. CAT-Seg[1]?
- The fine-tuning of the encoders for cost-based methods[1,2,3] would fundamentally differ from X-Decoder as these models fine-tune the encoder solely on the segmentation dataset, whereas X-Decoder maintain the image-caption objective during its training.
- Considering that these cost-based methods are able to avoid catastrophic forgetting while directly fine-tuning the encoders of CLIP, it would be interesting to observe how the effects of ConOVS to be complementary in terms of preventing forgetting issues.

**[Q2]** Given that the retrained baseline is expected to underperform due to not having sufficient training iterations, how much could retraining improve when it has fully converged?

**Ethical Concerns:**

["NO or VERY MINOR ethics concerns only"]

**Final Justification:**

The authors have provided detailed response, which resolved my concerns. Therefore, I maintain my positive rating.

**Limitations:**

Yes.

**Paper Formatting Concerns:**

No, there are no concerns regarding the paper format.

**Quality:**

3

**Strengths And Weaknesses:**

**Strengths**

**[S1]** The paper presents a well-founded motivation for incremental learning on the open-vocabulary setup, and constructs solid baselines for comparison, with the proposed method showing solid gains over the baseline.

**[S2]** The proposed solution is simple, yet sensible for solving the task. Furthermore, while it may be trivial for a MoE-based approach to prevent catastrophic forgetting, it is impressive to observe gains in unseen classes, where none of the "experts" should excel at.

**[S3]** The paper is generally well-written with helpful figures and to understand the framework, as well as providing solid experiments to validate the choices for estimating the interpolation factor.

**Weaknesses**

**[W1]** Some of the results, especially regarding the gains for unseen classes, are not well justified.
- While it is impressive to see the model improve on unseen classes, it is also counterintuitive to see the combined model to improve in such classes as none of the base models would be effective in identifying unseen classes. While the authors claim that the proposed method is generalizable, it is not explained where the gains are actually coming from.

**[W2]** Some of the design choices are not well justified.
- The method decides to freeze the encoder while only fine-tuning the decoder, without much justification or ablations. This can further relate to examining the applicability to streamline of works that focuses in fine-tuning the encoder of CLIP[1,2,3], which tend to show much better performance to its frozen counterparts, e.g. fc-clip.

[1] Cho S, Shin H, Hong S, Arnab A, Seo PH, Kim S. Cat-seg: Cost aggregation for open-vocabulary semantic segmentation. InProceedings of the IEEE/CVF Conference on Computer Vision and Pattern Recognition 2024 (pp. 4113-4123).

[2] Xie B, Cao J, Xie J, Khan FS, Pang Y. Sed: A simple encoder-decoder for open-vocabulary semantic segmentation. InProceedings of the IEEE/CVF conference on computer vision and pattern recognition 2024 (pp. 3426-3436).

[3] Lee M, Cho S, Lee J, Yang S, Choi H, Kim IJ, Lee S. Effective SAM Combination for Open-Vocabulary Semantic Segmentation. InProceedings of the Computer Vision and Pattern Recognition Conference 2025 (pp. 26081-26090).

---

> ### Author Rebuttal · Authors · 2025-07-31
>
> Thank you for your constructive review and for recognizing the motivation, simplicity, and effectiveness of our approach. We address your concerns in detail below.
>
> ---
>
> ### **Q1) The gains for unseen classes are not well justified. Further explanation or analysis is needed to clarify where these improvements come from.**
>
> We agree with the reviewer that further analysis is needed to justify the performance gains on unseen classes.
>
> To investigate the source of these improvements, we analyzed the performance on the zero-shot dataset ADE20K by comparing the PQ, SQ, and RQ scores of the baseline and our proposed method. Note that this experiment was conducted under Scenario 1, where the incremental dataset is Cityscapes.
>
> As shown in the table below, applying ConOVS to the baseline model (fc-clip) improves both PQ and RQ. However, the most significant gain is observed in SQ, which measures the quality of the predicted segmentation masks. This suggests that the improvements in unseen classes primarily stem from enhanced segmentation quality rather than better mask classification.
>
> |  | PQ | SQ | RQ |
> | :---- | :---- | :---- | :---- |
> | fc-clip | 23.5 | 61.7 | 28.3 |
> | \+ ConOVS (Ours) | **25.9 (+2.4)** | **73.1 (+11.4)** | **31.2 (+2.9)** |
>
> We will include this analysis in Section 6.1 (*Main Results*). Thank you for the feedback.
>
> ---
>
> ### **Q2) The choice to freeze the encoder and only fine-tune the decoder is not well justified. An ablation on this design choice is needed.**
>
> We agree with the reviewer that, given recent works that fine-tune the CLIP encoder \[1,2\], a more detailed ablation study on our design choice to freeze the encoder and only fine-tune the decoder is necessary.
>
> To this end, we experimented with three encoder fine-tuning strategies: 1\) fine-tuning only the last block of the encoder,  2\) fine-tuning only the LayerNorm modules, and  3\) replacing all MLP layers with LoRA modules and fine-tuning them. Note that, unlike prior attention-based methods designed for ViT-style CLIP encoders \[1,2\], our strategies are tailored for ConvNeXt-based encoders used in fc-clip.
>
> As shown in the table below, all three encoder fine-tuning strategies yield lower performance on the incremental dataset compared to our design choice of fine-tuning only the decoder. We attribute this to the architectural difference: while previous methods \[1,2\] generate segmentation masks directly from the encoder and benefit from encoder fine-tuning, fc-clip generates masks in the decoder's mask head. This suggests that decoder fine-tuning is more effective for improving segmentation performance in our setup.
>
> |  | coco (pre-training) | cityscapes (incremental) | ade20k (zero-shot) |
> | :---- | :---- | :---- | :---- |
> | fc-clip | 50.1 | 44.0 | 23.5 |
> | Last Block | 50.0 | 50.4 | 25.8 |
> | LayerNorm | 49.9 | 44.0 | 25.8 |
> | LoRA | **50.7** | 47.9 | 25.9 |
> | Only decoder | 50.4 | **64.4** | **26.0** |
>
> We will include this analysis in Section 6.2 (*Ablation Study*).
>
> > \[1\] Cho, Seokju, et al. *Cat-seg: Cost aggregation for open-vocabulary semantic segmentation. CVPR* 2024\.
> \[2\] Lee, Minhyeok, et al. *Effective SAM Combination for Open-Vocabulary Semantic Segmentation.* CVPR 2025\.
>
> ---
>
> ### **Q3) Furthermore, could the proposed method be applicable to cost-based OVS methods that fine-tune the encoder, e.g. CAT-Seg?**
>
> In general, our method is also applicable to cost-based OVS approaches. Designed to merge independently trained models, it remains compatible with techniques that fine-tune the CLIP encoder. While cost-based methods typically generate segmentation maps by post-processing encoder features, our method does not rely on such steps, making it directly applicable without modification.
>
> However, when applying our method to cost-based OVS models, the encoding process must be performed twice. This is because our method computes the interpolation factor based on the proximity of the input sample, which requires an initial forward pass through the encoder. After determining the interpolation factor, a second forward pass must be performed using the merged encoder to generate the final feature representation. As a result, an increase in inference time is expected.
>
> We believe that extending our method to encoder fine-tuning frameworks, such as cost-based OVS models, is an interesting direction for future work. We will include this discussion in the newly added *Future Works* section following Section 7\.
>
> ---
>
> ### **Q4) Given that cost-based methods already mitigate forgetting by fine-tuning CLIP encoders, it would be interesting to see whether ConOVS provides complementary benefits.**
>
> Our method is complementary to cost-based OVS models in terms of preventing catastrophic forgetting. When our method is applied to cost-based OVS techniques, a cost-based expert is trained for each domain. The merging of these experts is then guided by interpolation factors computed using the original CLIP encoder, which remains accessible at all times. As a result, we expect that combining our method with cost-based OVS models can further enhance their inherent ability to mitigate catastrophic forgetting.
>
> We consider empirically verifying this possibility an interesting direction for future work, and we plan to include this discussion in the *Future Work* section of the final version.
>
> ---
>
> ### **Q5) Given that the current retraining baseline underperforms due to limited training, how much improvement can be expected if it is allowed to fully converge?**
>
> We agree that evaluating the retraining baseline with longer training iterations offers valuable insights into the continual learning scenario for OVS. To conduct this analysis, we performed an experiment using Cityscapes as the incremental dataset.
>
> As shown in the table below, retraining the model with a longer training schedule (100k iterations) leads to better performance than our method on the pre-training and incremental datasets.
>
> | method | training iterations | coco (pretraining) | cityscapes (incremental) | ade20k (zero-shot) |
> | :---- | :---- | :---- | :---- | :---- |
> | retraining | 10k | 50.7 | 61.9 | 25.2 |
> | retraining | 20k | 50.7 | 62.5 | 25.4 |
> | retraining | 40k | **51.0** | 63.3 | 25.3 |
> | retraining | 100k | **51.0** | **64.4** | 25.5 |
> | ConOVS (Ours) | 10k | 50.4 | 64.2 | **26.0** |
>
> However, despite consuming significantly more computational resources, the retraining method does not yield substantially better performance than ConOVS. One possible explanation for the limited performance of retraining is task interference \[3,4\], which might occur when training on multiple domains simultaneously. In contrast, ConOVS avoids this issue by training domain-specific experts independently and combining them adaptively at inference time. As a result, our method achieves comparable performance than retraining, while requiring significantly less computational cost.
>
> We will include this analysis in the final version of the paper. We appreciate the helpful feedback.
>
> > \[3\] Yu, Tianhe, et al. *Gradient surgery for multi-task learning.* NeurIPS 2020\.
> \[4\] Chen, Zhao, et al. *Gradnorm: Gradient normalization for adaptive loss balancing in deep multitask networks.* ICML 2018\.

---

> ### Comment · Reviewer_5TFJ · 2025-08-04
>
> Thank you for the detailed response. The rebuttals have resolved my concerns, and it is nice to see results on additional datasets presented on responses to other reviews. I also agree on what authors raised as consideration for finetuning the encoder, and seems sound to leave it out for future work.

---

> > ### Author Response · Authors · 2025-08-06
> >
> > Dear Reviewer 5TFJ,
> >
> >
> > Thank you for your constructive feedback on our paper. The points you raised are insightful and have been valuable in improving our paper.
> >
> > We would also like to kindly remind you that your review rating does not appear to have been updated yet.
> > We would greatly appreciate it if you could take a moment to update it at your convenience.
> >
> >
> > Best regards,
> > The Authors

---

### Official Review · Reviewer_yU8U · 2025-07-02

**Clarity:** 3
**Significance:** 3
**Originality:** 3
**Rating:** 4
**Confidence:** 3

**Summary:**

This paper proposes a continuous learning method, ConOVS, to address the challenges of Open Vocabulary Segmentation (OVS) models in sequential data collection scenarios. ConOVS based on the MoE framework, achieves dynamic adaptation through the following mechanisms:
- Training phase: Fine-tune the decoder for each new dataset, generate independent experts, and calculate the multivariate normal distribution (MVN) of the image/text embeddings.
- Inference stage: Generate interpolation factors based on the similarity between the input samples and the distribution of each dataset (through MVN probability estimation), and dynamically combine the weights of the expert decoder.
The effectiveness of ConOVS was verified in three scenarios (COCO pre-training + 20K incremental data from Cityscapes/ADE) and multiple zero-shot datasets (such as LVIS, BDD100K). The results show that it outperforms the baseline method on pre-trained, incremental and zero-shot datasets, and alleviates the problem of catastrophic forgetting.

**Questions:**

see weakness

**Ethical Concerns:**

["Major Concern: Improper research involving human subjects"]

**Final Justification:**

The author's response solves my concern. I keep my rating as 4: Borderline accept.

**Limitations:**

yes

**Quality:**

3

**Strengths And Weaknesses:**

Strength:

- Experiment:
The segmentation performance was evaluated on over 10 datasets, covering multiple scenarios (single/multiple incremental datasets) and models (fc-clip, X-Decoder). The performance shows the enhance in performance.

- In-depth analysis:
The ablation experiment verified the key designs (such as the MVN distribution being superior to K-means/KDE, and the image + text modal combination being superior to the single modal).

Weakness：
- Although ConOVS claims to reduce retraining costs, the paper lacks a demonstration on the training cost comparison betwing retraining / SFT and continuous learning. Maybe detailed training time / resource / computational cost analysis is needed.

- The experimental setup relies primarily on only two or three incremental datasets (Cityscapes and ADE20K). The lack of evaluation under longer learning sequences (e.g., involving 5 to 10 incremental datasets) leaves open questions regarding the scalability and robustness of the method in handling high diversity and complex domain shifts over time.

---

> ### Author Rebuttal · Authors · 2025-07-31
>
> Thank you for highlighting the strengths of our work, particularly the thorough experimental validation and in-depth analysis. Below, we provide detailed responses to your concerns.
>
> ---
>
> ### **Q1) The paper lacks a comparison of training costs between retraining, SFT, and continual learning. A detailed analysis of time and resource usage is needed.**
>
> As the reviewer mentioned, the original draft did not include an explicit comparison of training costs across methods, which we acknowledge as a critical omission. Although our intention was to demonstrate the efficiency of ConOVS by showing that it achieves better performance than retraining with similar computational cost, this point was not sufficiently supported without the training cost comparison. Thank you for highlighting this issue.
>
> To address this, we measured the training time and GPU memory usage required by each method and summarized the results in the table below. Note that all values are reported as relative figures normalized to the computational cost of standard fine-tuning.
>
> | Method         | Scenario 1 (Cityscapes) |       | Scenario 2 (ADE20K) |       |
> |:--------------:|:-----------------------:|:-----:|:-------------------:|:-----:|
> |                | Training Time           | GPU Memory | Training Time     | GPU Memory |
> | Fine-tuning    | 1.00                    | 1.00       | 1.00              | 1.00       |
> | Retraining     | 1.25                    | 0.86       | 1.40              | 0.95       |
> | ER             | 1.25                    | 0.86       | 1.40              | 0.95       |
> | LwF            | 1.28                    | 1.07       | 1.21              | 1.13       |
> | EWC            | 1.07                    | 1.00       | 1.01              | 1.03       |
> | ECLIPSE        | 0.75                    | 0.57       | 0.74              | 0.52       |
> | ConOVS (Ours)  | 1.00                    | 1.00       | 1.00              | 1.00       |
>
>
> According to this comparison, the training time does not vary significantly across methods. This is expected, as we fixed the number of training iterations to ensure a fair comparison. One exception is ECLIPSE, which shows lower training cost. This is due to its use of Visual Prompt Tuning, which updates only a small subset of model parameters. While this leads to higher training efficiency, our results indicate that ConOVS outperforms ECLIPSE in terms of overall segmentation performance. We attribute this performance gap to the inherent limitations of updating only a fraction of the model parameters.
>
> We will include this discussion and the updated table in Section 6 (*Experiments*) and Table 1\. We appreciate your thoughtful feedback.
>
> ---
>
> ### **Q2) The use of only 2–3 incremental datasets limits evaluation of scalability and robustness under longer learning sequences and diverse domain shifts.**
>
> We agree that evaluating our method with a larger number of incremental datasets is important for demonstrating generalizability. To address this, we designed a new scenario where the model is sequentially trained on five datasets: COCO, Cityscapes, ADE20K, BDD100K, and Mapillary Vistas. While we aimed to include more datasets, we were limited to those providing panoptic segmentation annotations.
>
> The results of this extended experiment are shown in the table below. As can be seen, our method outperforms the baseline, fine-tuning, and retraining approaches. These results indicate that our method remains effective as the number of incremental datasets increases, demonstrating its scalability under complex domain shifts.
>
> |  | pre-training | incremental | |  |  | zero-shot |  |  |  |  |  | Average |
> | :---- | :---: | :---: | :---: | :---: | :---: | :---: | :---: | :---: | :---: | :---: | :---: | :---: |
> |  | COCO | Cityscapes | A-150 | BDD100K | Mapillary | LVIS | PC-59 | PC-459 | PAS-20 | PAS-21 | A-847 |  |
> | fc-clip | 50.1 | 44.0 | 23.5 | 19.0 | 26.0 | 20.5 | 53.0 | 16.9 | 93.1 | 80.2 | 13.8 | 40.0 |
> | fine-tuning | 31.7 | 55.5 | 28.7 | 25.8 | 36.3 | 17.7 | 49.8 | 16.3 | 90.0 | 73.9 | 14.0 | 40.0 |
> | retraining | 49.0 | 62.3 | 36.3 | 28.2 | 34.5 | 21.7 | **53.6** | **17.4** | **94.0** | **80.7** | 15.6 | 44.8 |
> | ConOVS (Ours) | **50.1** | **62.9** | **38.5** | **29.1** | **35.0** | **21.8** | **53.6** | 17.3 | 93.2 | 80.4 | **16.0** | **45.3** |
>
> We will include this experiment and its discussion in Section 6.1 (*Main Results*). Thank you for the valuable feedback.

---

> > ### Comment · Reviewer_yU8U · 2025-08-05
> >
> > Thanks for the authors response, which solves my concern. For one more small suggestion, providing an absolute value about training time and GPU (what 1.0 actually stands for in Q1 table) may helps reader understand better about the task resource.

---

> > > ### Author Response · Authors · 2025-08-06
> > >
> > > Dear Reviewer yU8U,
> > >
> > >
> > > Thank you for your thoughtful feedback on our paper.
> > > The points you raised are insightful and contribute meaningfully to improving our paper.
> > >
> > > We also appreciate your suggestion to include the absolute values of training time and GPU memory usage. We will incorporate this information into the table caption. Thank you again for the helpful recommendation.
> > >
> > >
> > > Best regards,
> > >
> > > The Authors

---

### Official Review · Reviewer_hsbu · 2025-07-03

**Clarity:** 3
**Significance:** 3
**Originality:** 3
**Rating:** 5
**Confidence:** 4

**Summary:**

The paper proposes a continual learning framework for open-vocabulary segmentation (OVS), addressing the practical scenario where data arrives incrementally over time and the model needs to adapt to new datasets. The authors introduce a method that trains a separate mask decoder for each incoming dataset and combines these decoders in a Mixture-of-Experts (MoE) manner during inference. Compared to full model retraining, the proposed approach is more cost-effective. It also mitigates catastrophic forgetting more effectively than fine-tuning, and improves performance on both incremental and zero-shot test datasets—surpassing existing continual learning methods on OVS.

**Questions:**

These questions come from comments on weaknesses

- Could this approach also be applied to OVD?

- Why does continual learning yield better performance than retraining from scratch as shown in Table 1?

- The paper assumes that each incremental dataset belongs to a different domain, which may not hold true in all scenarios. What happens if we do not know the domain of each incoming dataset? For example, what if the incoming data at a given time step is a mixture of multiple domains (e.g., ADE20K and Cityscapes)? How robust is the method in such situations?

- If an incremental dataset is small or weakly labeled, how does that affect model performance? Will the proposed approach still perform well in such low-resource settings?

**Ethical Concerns:**

["NO or VERY MINOR ethics concerns only"]

**Final Justification:**

The rebuttal has further clarified and provided more insights to my questions. It's a good paper and I'm fine to raise my rating.

**Limitations:**

Yes

**Quality:**

3

**Strengths And Weaknesses:**

Strengths:

- While other OVS methods focus on training a segmentation model once and applying it everywhere, this paper addresses the scenario where data arrives incrementally over time and adapts the OVS model using the new incremental data. This paradigm has been shown to be effective for both incremental and zero-shot dataset performance. The observation aligns with prior work such as [1], where incorporating additional data during training also led to performance improvements in OVS.

- The integration of continual learning with a Mixture-of-Experts (MoE) strategy for OVS is particularly novel and well-motivated. A standard OVS model typically consists of three components: a text encoder, an image encoder, and a mask decoder. This work proposes learning a separate mask decoder for each incoming dataset and combining their weights during inference. This design is shown to be highly effective, as demonstrated in Tables 1, 2, and 3.

- The paper is clearly written and well-organized, making it easy to follow the motivation, methodology, and results.

[1] Scaling Open-Vocabulary Image Segmentation with Image-Level Labels

Weaknesses:
- The framework seems generalizable to many tasks, and the paper focuses only on OVS. However, there is no specific design for segmentation. A closely related line of work is Open-Vocabulary Detection (OVD). Could this approach also be applied
to OVD?

- As shown in Table 1, the continual learning model outperforms the retrained model. Why does continual learning yield better performance than retraining from scratch?

- The paper assumes that each incremental dataset belongs to a different domain, which may not hold true in all scenarios. What happens if we do not know the domain of each incoming dataset? For example, what if the incoming data at a given time step is a mixture of multiple domains (e.g., ADE20K and Cityscapes)? How robust is the method in such situations?

- If an incremental dataset is small or weakly labeled, how does that affect model performance? Will the proposed approach still perform well in such low-resource settings?

---

> ### Author Rebuttal · Authors · 2025-07-31
>
> Thank you for your thoughtful review and for highlighting the novelty and clarity of our approach. We address your questions regarding generalizability and robustness in detail below.
>
> ---
>
> ### **Q1) Could this approach also be applied to OVD?**
>
> In general, our proposed method is not limited to open-vocabulary segmentation (OVS) but can also be extended to open-vocabulary detection (OVD) frameworks.
>
> Our approach is applicable to models with an encoder-decoder architecture that perform classification based on the similarity between image and text embeddings. This structural characteristic is shared by most OVD methods. For instance, YOLO-World \[1\] adopts a modular design consisting of an encoder and a prediction head, and omits the use of a conventional fc-based classifier. Given this structure, all components of our method can be directly incorporated without modification.
>
> To be more specific, for each newly introduced training dataset, an MVN distribution can be formed using the encoder's embeddings. During inference, the interpolation weights can be dynamically adjusted based on the proximity between the input sample and each domain, enabling the model to incrementally extend its recognition capability.
>
> We believe that applying our method to OVD models represents a promising direction for future work. Accordingly, we plan to include this discussion in the newly added *Future Work* section following Section 7 (*Limitation*).
>
> > \[1\] Cheng, Tianheng, et al. *YOLO-World: Real-time Open-Vocabulary Object Detection.* CVPR 2024\.
>
> ---
>
> ### **Q2) Why does continual learning yield better performance than retraining from scratch, as shown in Table 1?**
>
> In our experiments, the retraining method showed relatively lower performance because we conducted all methods under the same computational budget. Since retraining involves learning from a significantly larger amount of data compared to our method, it requires a longer training schedule to reach convergence.
>
> To analyze this more precisely, we conducted additional experiments where the retraining method was trained for longer durations. As shown in the table below, when trained with a sufficiently long schedule (100k iterations), the retraining method outperforms our method on the pretraining and incremental datasets.
>
> | method | training iterations | coco (pretraining) | cityscapes (incremental) | ade20k (zero-shot) |
> | :---- | :---- | :---- | :---- | :---- |
> | retraining | 10k | 50.7 | 61.9 | 25.2 |
> | retraining | 20k | 50.7 | 62.5 | 25.4 |
> | retraining | 40k | **51.0** | 63.3 | 25.3 |
> | retraining | 100k | **51.0** | **64.4** | 25.5 |
> | ConOVS (Ours) | 10k | 50.4 | 64.2 | **26.0** |
>
> However, despite consuming significantly more computational resources, the retraining method does not yield substantially better performance than ConOVS. One possible explanation for the limited performance of retraining is task interference \[2,3\], which might occur when training on multiple domains simultaneously. In contrast, ConOVS avoids this issue by training domain-specific experts independently and combining them adaptively at inference time. As a result, our method achieves comparable performance than retraining, while requiring significantly less computational cost.
>
> Additionally, we acknowledge that Table 1 in the original manuscript does not clearly convey how the training cost of retraining compares with other methods. To address this, we have included a comparison of training time and GPU memory usage across all methods. Please refer to our response to Reviewer yU8U’s Q1 for more details.
>
> We recognize that this important explanation was not included in the original draft and have now added the relevant discussion to Section 6.1 (*Main Results*) and the caption of Table 1\. We appreciate the constructive feedback.
>
> > \[2\] Yu, Tianhe, et al. *Gradient surgery for multi-task learning.*, NeurIPS 2020\.
> \[3\] Chen, Zhao, et al. *Gradnorm: Gradient normalization for adaptive loss balancing in deep multitask networks.*, ICML 2018\.
>
> ### **Q3) The method assumes each incremental dataset is from a distinct domain, which may not always be the case. How does it handle mixed or unknown domains during training?**
>
> As suggested by the reviewer, we conducted an additional experiment to examine whether our method remains effective when a single incremental dataset consists of multiple domains. To this end, we combined Cityscapes and ADE20K into a single incremental dataset. Note that the performance on the zero-shot dataset was computed by averaging the results from four datasets: PC-59, PC-459, PAS-20, and PAS-21.
>
> As shown in the table below, our method preserves the performance of the baseline fc-clip on the pre-training dataset, while improving performance on both the incremental and zero-shot datasets. This demonstrates that our method remains robust and effective even when the incremental dataset includes samples from diverse domains.
>
> | method | coco (pretraining) | cityscapes (incremental) | ade20k (incremental) | Average on 4 zero-shot datasets |
> | :---- | :---- | :---- | :---- | :---- |
> | fc-clip | 50.1 | 44.0 | 23.5 | 60.8 |
> | ConOVS (ours) | **50.1** | **60.4** | **31.5** | **61.3** |
>
> We will include this analysis in the final version of the paper. We appreciate the helpful feedback.
>
> ---
>
> ### **Q4) How does the method perform when the incremental dataset is small or weakly labeled? Is it robust in low-resource settings?**
>
> To examine whether the proposed method remains effective in low-resource settings, we conducted an additional experiment using a small incremental dataset. Specifically, we randomly sampled only 5% of the Cityscapes training dataset and used it as the incremental dataset.
>
> As shown in the table below, our method successfully maintains the performance on the pre-training dataset while improving performance on both the incremental and zero-shot datasets, even when trained with the small incremental dataset. This demonstrates that the proposed method remains effective in low-resource settings where the number of incremental samples is limited.
>
> | Method | COCO (pre-training) | Cityscapes (incremental) | ADE20K (zero-shot) |
> | :---- | :---- | :---- | :---- |
> | fc-clip | 50.1 | 44.0 | 23.5 |
> | ConOVS (Ours) | 50.0 | **48.8** | **25.9** |
>
> We also considered evaluating our approach under weakly labeled incremental settings. However, due to the lack of publicly available datasets that contain appropriate weak annotations, such as pseudo labels for panoptic segmentation, we were unable to conduct this experiment.
>
> Nevertheless, we believe that our method can also be applied in weakly labeled settings. In fact, we consider our approach to be effective in overcoming the common issue of low-quality segmentation annotations found in weakly labeled datasets. This is because our technique constructs the MVN distribution using only image and text embeddings, without relying on segmentation annotations. Therefore, we believe there are no constraints in forming the MVN distribution, and we expect to accurately estimate the interpolation coefficients. While label noise may affect the overall performance, we believe our method can still contribute to effectively enhancing the model’s recognition ability even in weakly labeled environments.
>
> We will include the experiment and analysis regarding small incremental datasets in Section 6 (*Experiments*), and the discussion on weakly labeled datasets will be added to the *Future Work* section. We appreciate your thoughtful feedback.

---

### Note · Authors · 2025-08-13

To the Reviewers and Area Chairs,


We sincerely thank the reviewers and area chairs for the time and effort you dedicated to evaluating our paper. We also appreciate your attention to our rebuttal during the discussion period.

Your feedback is very helpful in improving the quality of our work.


Best regards,

The Authors

---

### Decision · Program_Chairs · 2025-09-17

**Decision:**

Accept (poster)

**Comment:**

This work presents a continual learning framework for open-vocabulary segmentation, where data is presented in a sequential manner. Reviewers found that the proposed strategy is novel and well-motivated, which is empirically supported by extensive experiments across multiple datasets and models. Furthermore, the impact of the different components was properly evaluated on further ablation studies. Despite these strengths, reviewers also pointed to several weaknesses (e.g., several results or choices not properly justified, lack of detailed computational overhead analysis, as well as further clarification), which were successfully addressed by the authors during the rebuttal/discussion process. Considering all these points I recommend the acceptance of this work.